# Intra-Articular Injections of Autologous Adipose Tissue or Platelet-Rich Plasma Comparably Improve Clinical and Functional Outcomes in Patients with Knee Osteoarthritis

**DOI:** 10.3390/biomedicines10030684

**Published:** 2022-03-16

**Authors:** Jakub Kaszyński, Paweł Bąkowski, Bartosz Kiedrowski, Łukasz Stołowski, Anna Wasilewska-Burczyk, Kamilla Grzywacz, Tomasz Piontek

**Affiliations:** 1Department of Orthopedic Surgery, Rehasport Clinic, 60-201 Poznań, Poland; jakub.kaszynski@rehasport.pl (J.K.); bartosz.kiedrowski@rehasport.pl (B.K.); lukasz.stolowski@rehasport.pl (Ł.S.); tomasz.piontek@rehasport.pl (T.P.); 2Institute of Bioorganic Chemistry Polish Academy of Sciences, 61-704 Poznań, Poland; awasilewska@ibch.poznan.pl (A.W.-B.); kgrzywacz@ibch.poznan.pl (K.G.); 3Department of Spine Disorders and Pediatric Orthopedics, University of Medical Sciences Poznań, 61-701 Poznań, Poland

**Keywords:** adipose-derived stem cells, autologous adipose tissue, platelet-rich plasma, knee osteoarthritis

## Abstract

The use of biologic therapies for the management of knee osteoarthritis (OA) has largely increased in recent years. The purpose of this study was to evaluate the efficiency and the therapeutic potential of platelet-rich plasma (PRP) and autologous adipose tissue (AAT) injections as a treatment for knee OA. Sixty participants were enrolled in the study: 20 healthy ones and 40 with minimal to moderate knee OA (KL I-III). The OA patients were randomly assigned either to the PRP or to the AAT group. The PRP samples showed a low expression level of NF-κB-responsive gene CCL5 and high expression levels of classic inflammatory and TNF-l INF responses. The AAT injection product was prepared using a Lipogems device, and its regenerative potential as well as the ability for expansion of mesenchymal stem cells were tested in the cell culture conditions. The patient assessments were carried out five times. Significant improvement was observed regardless of the treatment method in the VAS, KOOS, WOMAC and IKDC 2000 subjective evaluations as well as in the functional parameters. Intra-articular injections of AAT or PRP improved pain, symptoms, quality of life and functional capacity with a comparable effectiveness in the patients with mild to moderate knee osteoarthritis.

## 1. Introduction

Knee osteoarthritis (OA) is a debilitating disease causing persistent pain, joint effusion and affecting the quality of life of people struggling with this disease. OA leads to irreversible changes in the articular cartilage and the subchondral bone, osteophytes formation and changes in the knee axial alignment [1,2]. There is no one universal cure for OA. Non-operative rehabilitation, including the range of motion restoration, strengthening of the quadriceps, hamstrings and gluteal muscles and a manual therapy combined with weight loss and patient education, are the first line recommendations for patients with an early-stage disease. The topical and oral non-steroidal anti-inflammatory drugs (NSAIDs) are helpful in pain alleviation. If a conservative treatment fails, a more invasive treatment may be considered. The operative treatment includes: arthroscopy, cartilage repair, osteotomy or knee arthroplasty [1,2,3,4,5,6,7].

The most challenging aspect of the knee osteoarthritis treatment is a degeneration of the articular cartilage because of its limited potential for regeneration. Therefore, the biological therapies aimed at regeneration stimulation progressively gain more attention among orthopedists.

The platelet-rich plasma (PRP) is one of the commonly used biological therapies in orthopedics. An indisputable advantage of this treatment involves its safety and minimal invasiveness. PRP is an autologous platelet concentrate, prepared from the patient’s whole blood by deprivation of the red blood cells. The platelets collected in the PRP are activated to release the growth factors (GFs) and the cytokines. The primary GFs in PRP include: a basic fibroblast growth factor (FGF-2), a platelet-derived growth factor (PDGF), a profibrotic transforming growth factor beta, which is a critical mediator of the epithelial to mesenchymal cell transition (TGF-β), a vascular endothelial growth factor (VEGF) and an insulin-like growth factor (IGF). The growth factors in the PRP possess regenerative potential via their anti-inflammatory, chemotactic, antiapoptotic and proliferative effects on the fibroblasts [8,9]. The cytokines mediate the inflammatory processes and promote tissue healing [10]. The cytokines present in the PRP include: an interleukin-1 (IL-1β), a tumor necrosis factor-α (TNF-α) and matrix metalloproteinases (e.g., MMP-9, MMP-13). The effectiveness of the PRP has been evaluated in many studies [11,12,13,14,15].

The use of the autologous adipose tissue (AAT) as a source of the mesenchymal stem cells (MSCs) has also been recognized in recent orthopedic treatments [1,16,17,18,19,20,21,22,23]. Mesenchymal stem cells are multipotent and exert anti-inflammatory and immunomodulatory effects. They can differentiate into multiple cell types, including osteoblasts or chondrocytes [24,25]. The cytokines secreted from the MSCs include: a prostaglandin E (PGE2), a granulocyte-macrophage colony-stimulating factor (GM-CSF) and a multitude of interleukins (IL), e.g., IL-1RA, IL-7, IL-8, IL-10 and IL-11 [26,27]. The adipose-derived stem cells (ASCs) are a subset of mesenchymal stem cells that can be obtained easily from the subcutaneous adipose tissues during a minimally invasive procedure.

Intra-articularly injected PRP and AAT were hypothesized to promote cartilage repair and prevent or slow articular cartilage degeneration during OA development. There are multiple studies concentrated on the influence of the biological treatment on cartilage regeneration. However, there are only limited studies aimed at inspecting the effectiveness in improving patients’ function. Moreover, to our knowledge, no study aimed at a comparison of the effectiveness of the PRP and AAT biological treatments in patients suffering from knee osteoarthritis has been performed. In multiple studies [19,20,21,22,23], the patient-related outcomes measures were limited to the diagnostic imaging and subjective questionnaires without evaluating the patients’ functional capacity, which should stand as a primary aim of OA treatment. Therefore, the purpose of this study was to compare the clinical changes and the functional capacity in patients affected with knee osteoarthritis and treated with intra-articular injections of the autologous adipose tissue or the platelet-rich plasma.

## 2. Materials and Methods

### 2.1. Patient Selection

The study protocol for this randomized, controlled trial has already been published, in which the inclusion and exclusion criteria were explained in detail [28]. Briefly, the inclusion criteria consisted of: a symptomatic knee OA, the age between 45 and 65 years old, Kellgren–Lawrence OA grades from weak to moderate (KL I–III), no or minimal positive effects of previous conservative treatment, the minimum visual analogue scale (VAS) pain score of 4 and the VAS pain score equal or less than 2 in the contralateral knee. The exclusion criteria included the use of local corticosteroids up to 3 months or hyaluronic acid injections up to 6 months before the treatment, joint infections (present and past), previous knee arthroscopy surgery (up to 1 year), peripheral inflammatory diseases (e.g., rheumatoid arthritis, spondyloarthropathies), total arthroplasty and osteotomy, joint ankylosis, dermatitis or dermatological disease at the intended injection site, coexistence of degenerative changes in other limb joints (hip, foot), cancer, oral corticosteroid therapy, use of medicines that affect blood clotting, pregnancy or breast-feeding.

In order to eliminate pre-analytical differences that might bias the study, the randomization process of the patients’ assignment to the particular treatment group was performed. After the qualification procedure, the participants were asked by the physician to draw a card with a number: 1 or 2. When number 1 was drawn, the participant was assigned to the AAT group, number 2—to the PRP group.

Blind methodology was used to prevent the possible bias in the subjective evaluation derived from the knowledge of patient’s group allocation. The personnel dedicated to the evaluation of the responses to the treatment did not have information regarding group allocation. Initially, 74 participants were enrolled for the study: 20 healthy volunteers and 54 patients with symptomatic knee OA. 54 OA participants eligible for the study were randomly allocated into the treatment groups: 28 patients received multiple PRP injections and 26 patients received a single AAT injection. Participants were assessed five times: before the treatment and 1, 3, 6 and 12 months after the treatment. During the 12-month follow-up time, 14 subjects were lost: 8 from the PRP group and 6 from the AAT group. Finally, in total, 60 patients were enrolled in the study: 20 patients in the PRP group, 20 patients in the AAT group and 20 healthy participants (control group).

### 2.2. PRP Preparation and Platelet Count

Platelet-rich plasma was prepared as already described [28]. Briefly, 40-mL peripheral blood sample was collected from each patient to the sterile collecting tubes containing anticoagulant. The blood was centrifuged at 2320× *g* for 7 min. at room temperature and the PRP was collected in a new, sterile tube. Platelet concentrations from the whole blood and PRP were measured using a hematology analyzer.

### 2.3. Lipoaspiration Procedure and AAT Manipulation

The lipoaspiration procedure and AAT manipulation was performed as already described [28]. Briefly, the lipoaspiration took place in the operating room under general anesthesia. Around 100 mL of the adipose tissue was harvested and processed with the Lipogems device (Lipogems, Milan, Italy). It is a closed, full-immersion, low pressure cylindrical system to produce a uniform product containing concentrated populations of adipocyte-derived mesenchymal stem cells. The final Lipogems product was collected into the new, sterile tubes.

Approximately 20 mg of the adipose tissue from each lipoaspiration (before Lipogems procedure) was fixed in 10% buffered formalin and embedded in paraffin. 4-µm-thick paraffin sections were used for histological analysis and were stained with hematoxylin and eosin. The results were observed under a light microscope.

A portion of 1.5 mL of the final Lipogems product was immediately immersed in 30 mL of the Dulbecco’s Modified Eagle Medium (DMEM) supplemented 1% penicillin-streptomycin and 1% bovine serum albumin (BSA) and transported to the laboratory with cooling. Next, the Lipogems product was slowly thawed at 37 °C and seeded in a T-75 cell culture flask in the minimum essential α-MEM medium supplemented with 20% heat-inactivated fetal bovine serum (FBS), antibiotics (200 units/mL penicillin, 100 µg/mL streptomycin) and l-glutamine (1%). The cell culture was incubated at 37 °C in a humidified atmosphere with 5% CO_2_. The adipocyte-derived mesenchymal cells were inspected under a light microscope.

### 2.4. Gene Expression Analysis

Total RNA was extracted from the platelet-rich plasma samples using Trizol, and 100 ng was reverse-transcribed into the cDNA in a 20-µL reaction volume with SuperScript IV Reverse Transcriptase (Invitrogen, Waltham, MA, USA) and random hexamer primers (Invitrogen). To assess the gene expression, 2 µL of cDNA were used for real-time PCR performed with a Mx3000P qPCR system system (Agilent, Santa Clara, CA, USA) and with the 5× HOT FIREPol EvaGreen qPCR Mix Plus kit (Solis Biodyne, Tartu, Estonia) following the manufacturer’s instructions.

Primers (5 µM) were used to evaluate the expression levels of interleukin 23A (IL23A), interleukin 1β (IL-1β), C-X-C Motif Chemokine Ligand 1 (CXCL1), C-X-C Motif Chemokine Ligand 3 (CXCL3), C-Motif for Chemokine Ligand 5 (CCL5), Matrix Metallopeptidase 3 (MMP3). Data were normalized using glyceraldehyde 3-phosphate dehydrogenase (GAPDH). Samples were run in triplicate, and the average threshold cycle (Ct) value was used for calculations. The relative quantification of the mRNA expression was calculated with the comparative Ct. The sequences of the primers are shown in Appendix A.

### 2.5. Injection Procedures

The joint injection was performed by two orthopedists (PB and TP). The PRP group received one cycle of the intra-articular PRP injection in the affected knee. A cycle consisted of three injections, given a half month apart. The AAT group received one dose of the final Lipogems product. All injections were performed in the same manner. Briefly, the patient was placed in a supine position with the affected knee extended, the orthopedist inserted a needle into the suprapatellar pouch and administrated the AAT or the PRP product.

After the treatment, the patients in both groups were allowed for partial weight-bearing for the first two weeks. After that, full-weight loading was allowed and patients were instructed to perform physical exercises.

### 2.6. The Subjective Evaluations

The subjective evaluations were performed using the Visual Analog Scale (VAS) and 4 self-evaluation questionnaires: the Knee Osteoarthritis Outcome Score (KOOS) [29], the Western Ontario and McMaster Universities Osteoarthritis Index (WOMAC) [30], the International Knee Documentation Committee 2000 (IKDC 2000) [31] and the Health Questionnaire EQ-5D-5L [32].

The visual analogue scale is used for determination of the pain intensity. The VAS scale is represented as a 10 cm-long line. The left side of the line (“0”) corresponds to “no pain” and the right side (“10”) represents “the worst pain ever”.

The Knee Osteoarthritis Outcome Score evaluates short- and long-term outcomes of the knee injury. It consists of 42 questions, grouped in 5 blocks: Pain, Other Symptoms, Function in daily living (ADL), Function in Sport and Recreation (Sport/Rec, and knee-related Quality of Life (QOL). The minimum score of “0” represents “extreme knee problems” and the maximum (“100”)—“no knee problems”.

The Western Ontario and McMaster Universities Osteoarthritis Index scale serves to measure the pain (score range “0”–“20”), the stiffness (“0”–“8”) and the functional limitations (“0”–“68”). The higher the score, the worse the results.

The International Knee Documentation Committee 2000 questionnaire holds 3 categories: symptoms, sports activity and knee function. Scores are scaled from 0 to 100. Higher scores represent higher levels of function.

The Health Questionnaire EQ-5D-5L consists of the descriptive scale and the VAS scale. The descriptive system is designed to assess the patient’s mobility, self-care, daily activities, pain/discomfort and anxiety/depression. The minimum score refers to “no problems” and the maximum—“extreme problems”. The VAS system records the patient’s self-rated health on a VAS scale, with “the best health you can imagine” and “the worst health you can imagine” endpoints.

### 2.7. The Functional Evaluations

The functional assessments were performed using the Timed Up and Go Test (TUG) [33], the 5 Times Sit to Stand Test (5 × STS) [34] and 10 m Walk Test (10 mWT) [35].

In the Timed Up and Go Test, the patient initially sits on a chair and then stands up and walks straight for 3 m. Next, the patient turns around and goes back to the starting position. The time of the exercise is measured precisely with a stopwatch.

During the 5 Times Sit to Stand Test, the patient sits on a chair without the back support with arms crossed on chest and then stands up to fully extend both hips and knees. After standing up, the patient sits down again and repeats the procedure five times. The result of this test is time of 5 exercises, measured with a stopwatch.

In the 10 m Walk Test, the starting position is when the patient stands and then walks straight for 10 m. The time of the 10 m walk is measured precisely with a stopwatch.

### 2.8. The Isometric Evaluations

The isometric evaluations were performed using the Maximal Voluntary Isometric Contraction (MVIC) measurement of extensors and flexors of the knee joint. MVIC measurement was performed using a Forcemeter FB 500 device (AXIS, Gdansk, Poland), dedicated for measuring the force during pressing and pulling activities. To perform the measurement, the patient initially sits on a bench with a measuring belt placed around the waist and above the ankle joint. The knee in the tested limb is flexed to 90 degrees. The procedure starts with the first extension of the knee, which lasts for 6 s. The force is measured in Newtons (N). The results are divided by the patient’s weight (kg) for data analysis.

### 2.9. Statistical Analysis

The sample size calculation showed that, in order to ensure the 99% confidence with a power of 80% (2-sided testing), a sample size of 18 participants is needed. Mean and standard deviation were used to present descriptive analysis, including demographics, BMI and stage of OA changes in Kellgren–Lawrence scale (KL). Shapiro–Wilk test was used to evaluate the data distribution. For further analysis, the *t*-test, the Mann–Whitney U test and the repeated measures ANOVA were used. Tukey test and NIR test were used for post hoc analysis. A *p*-value of < 0.05 was considered statistically significant.

### 2.10. Ethics

The study was approved by Bioethical Committee, Poznan University of Medical Sciences (no. 868/18). Written informed consent was obtained from all participants.

## 3. Results

### 3.1. Patients Characteristics

The demographic data of the patients are presented in Appendix A. The study included 60 participants: 40 patients randomly allocated into PRP or AAT groups and 20 healthy participants (control group). Twenty-three patients were characterized with a Kellgren–Lawrence (KL) grade of II (12 from the PRP group and 11 from the AAT group) and 17 patients—KL III. No significant differences in the demographic characteristics were found among the three groups.

### 3.2. PRP Characteristics

In order to test the regenerative potential of the PRP product, we conducted the analysis of its molecular composition. The platelet density was 204 ± 48 × 103/μL in the whole blood and 172 ± 17 × 104/μL in PRP. The platelet density in the PRP was 8.4 times that in the whole blood (*p* = 0.001), which is in agreement with previous results stating that the PRP should have at least five times the number of platelets compared to the whole blood to be considered as “platelet rich” [36].

The analyses of the RNA isolated from the PRP samples revealed that the PRP expressed a set of autologous growth factors and secretory proteins that may enhance the healing process on a cellular level (Figure 1). The PRP samples obtained from different patients expressed almost equal transcript levels. The PRP showed a low expression level of NF-κB-responsive gene CCL5 and high expression levels of classic inflammatory and TNF-l INF responses: IL23A (induces autoimmune inflammation), IL1-1β (a growth factor that stimulates the fibroblast proliferation), CXCL1 and 3 (growth factors participating in the inflammation) and a metallopeptidase MMP13 (a typical osteoblastic marker), as assumed [37].

### 3.3. AAT Characteristics

The histological analysis of the lipoaspirate is presented in Figure 2A. In order to test the regenerative potential of the Lipogems product used in the patients from the AAT group, we have tested its ability for the expansion of mesenchymal stem cells in the cell culture conditions. We have shown that the Lipogems product can be simply transferred into the tissue culture. MSCs slipped out from the tissue cluster product, starting after day 7, attached to the tissue culture plastic and reached 70 to 80% confluence in 28 days (Figure 2C).

### 3.4. The Subjective Evaluations

The subjective evaluations were performed using the VAS, KOOS, WOMAC, IKDC 2000 and EQ-5D-5L scales.

There were no statistically important differences in the VAS score values between the PRP and the AAT groups at any of the follow-ups, nor before the treatment (Appendix A). After 1 month, the VAS score in the PRP and AAT group decreased statistically when compared to the pre-treatment values (*p* = 0.019 and *p* = 0.037, respectively) and was constantly decreasing until the final follow-up. The VAS score achieved 12 months after the PRP or the AAT treatment was still significantly lower than for the healthy participants (*p* = 0.000 for both the PRP and AAT group). Already 1 month after the treatment, the WOMAC and IKDC 2000 scores in both groups and EQ-5D-5L in the PRP group increased statistically when compared to the pre-treatment values and were constantly increasing until the final follow-up (Figure 3). The increase in the WOMAC score was more prominent in the AAT than in the PRP group at 1 month follow-up (*p* = 0.030 and *p* = 0.000). A prominent increase in the WOMAC score was observed in the AAT group between the 6-months and 12-months follow-ups (*p* = 0.011). The differences in the IKDC 2000 scores observed for the AAT group between the following assessments were of the highest statistical importance. At the first follow-up (1 month after the treatment), the EQ-5D-5L score increased statistically when compared to the pre-treatment values only in the PRP group (*p* = 0.003) and was constantly increasing until the final follow-up. In the AAT group, the first differences in the EQ-5D-5L scores were observed 6 months after the treatment (*p* = 0.02, when compared to the pre-treatment values). The EQ-5D-5L scores comparable to those obtained by the control group were achieved already 1 month after the PRP or the AAT treatment. The highest WOMAC and IKDC 2000 score was achieved 12 months after the PRP or the AAT treatment and was still significantly lower than for the healthy participants (*p* = 0.000 for both PRP and the AAT group).

There were no statistically important differences in the KOOS score values between the PRP and the AAT groups at any of the follow-ups, nor before the treatment (Appendix A). Already 1 month after the treatment, the KOOS scores in both groups increased statistically when compared to the pre-treatment values; however, this increase was more prominent in the AAT group in all the subscales (Figure 4). The highest KOOS scores in all the subscales were achieved 12 months after the PRP or the AAT treatment, and these results were still significantly lower than for the healthy participants (*p* = 0.000 for both PRP and the AAT group in all the subscales).

### 3.5. The Functional Evaluations

The functional assessments were performed using the Timed Up and Go Test (TUG), the 5 Times Sit to Stand Test (5 × STS) and the 10 m Walk Test (10 mWT).

At the baseline, there was a significant difference between the results obtained by the AAT and the PRP groups in TUG (*p* = 0.021), 5 × STS (*p* = 0.017) and 10 mWT (*p* = 0.030) (Appendix A). Such differences persisted in the 5 × STS through the 1-month until the 3-months follow-up (*p* = 0.000 and *p* = 0.017, respectively). From 3M on, there was no significant difference between the treated groups. Both groups significantly improved over 12 months (Figure 5) regardless of the treatment method. However, an improvement greater than the Minimal Detectable Change (MDC) (MDC: TUG = 1.01 s; STS = 1.91 s; 10WT = 0.65 s) was achieved only by the AAT group, already 1 month after the treatment. The functional scores comparable to those obtained by the control group were achieved 3 months after the AAT treatment in TUG and 6 months after the treatment in 10 mWT. The lowest 5 × STS score was achieved 12 months after the AAT treatment but was still significantly higher than for the healthy participants (*p* = 0.002). The functional scores comparable to those obtained by the control group were achieved 1 month after the PRP treatment in TUG and 10 mWT and 3 months after the treatment in 5 × STS.

### 3.6. The Isometric Evaluations

Isometric evaluations were performed using the Maximal Voluntary Isometric Contraction (MVIC) measurements of the extensors and the flexors of the knee joint in relation to the body mass.

There was no significant difference at the baseline between the AAT and the PRP groups in MVIC of the knee extensors nor flexors. Both treatment groups improved over time in MVIC, and a significant difference between the tendency to improve the results in the PRP and the AAT group was observed 3 months after the treatment in MVIC of knee flexors (*p* = 0.02).

The patients treated with the AAT achieved a significant improvement in the MVIC of both extensors and flexors at 3-month follow-up when compared to the pretreatment values (*p* = 0.002) and patients from the PRP group—already after 1 month (0.04) (Appendix A and Figure 6). The isometric scores comparable to those obtained by the control group were achieved 6 months after the AAT treatment in the MVIC of extensors.

## 4. Discussion

This study demonstrated that two biological treatments, PRP and AAT injections, effectively improved the symptoms and the functional capacity in patients with mild to moderate knee osteoarthritis, reflecting the comparable therapeutic use of these two treatments.

In the available literature, there is no robust nor stringent evidence supported with randomized controlled trials for the effectiveness of the knee osteoarthritis treatment with intra-articular injections of AAT. However, several small sample size studies exist, which showed, as in our studies, that it might be a very effective form of a biological treatment [17,38,39]. All four studies evaluated the AAT injection results using subjective measurements solely. Barfod and Blond [17] included 20 patients suffering from knee OA into a study (KL I–IV). They evaluated the subjective outcomes of the AAT treatment with the KOOS scale at the baseline and 3, 6 and 12 months after the AAT treatment. The participants significantly improved in each KOOS subscale at the final follow-up, but the improvement in the “Other symptoms” subscale was not clinically significant. Two patients had to undergo an additional operative treatment. Moreover, they found that 15 out of the 20 patients would decide to receive this type of treatment again. Spasovski et al. [39] reported that, in a group of nine patients, the IA injections of AAT significantly improved the results of KOOS, Tegner-Lysholm and VAS Pain scores. Furthermore, they reported the improvement in MRI scans (2D Mocart score: 43.0 ± 7.2 at the baseline versus 63.0 ± 17.1 at 18-months follow-up) and a cartilage enhancement, but they did not observe any differences in X-ray results. On the other hand, Fodor and Paulseth [38] did not find any changes in the MRI scans in eight treated knee joints, but they evaluated the scans 3 months after treatment solely.

Some authors have tested the effectiveness of the AAT treatment on severe knee OA (KL III–IV) [40,41,42]. Lapuente et al. [40] included in his study 50 patients with bilateral knee OA; however, their final AAT injection product was prepared using an enzymatic digestion instead of the Lipogems device, like in this study. The subjective outcomes were measured with the Lequesne, WOMAC and VAS scales and the biochemical profile of the synovial fluid was assessed. They reported a significant improvement in each scoring system at 1-year follow-up regardless of the patient’s age or the severity of OA. The analysis of the synovial fluid showed a significant increase for anabolic and anti-inflammatory molecules (IGF-1, IL-10) and a decrease of pro-inflammatory, catabolic molecules (MMP-2, IL-1B, IL-6 and IL-8). Hudetz et al. [41] treated 20 patients with late-stage knee OA with micro-fragmented lipoaspirate. Three patients needed to undergo a total knee arthroplasty within a year after the AAT treatment. The remaining 17 patients showed a significant improvement in KOOS and WOMAC at 1-year follow-up.

Bone marrow edema visible on MRI scans at the baseline is thought to be a good outcome predictive factor for an improvement in the pain subscale in KOOS and WOMAC. The study of Yokota et al. [42] involved 13 patients with bilateral knee OA (KL III–IV), treated with an IA injection of autologous adipose tissue, prepared with an enzymatic digestion. They monitored the effects of the treatment over 6 months. They observed a significant improvement in the VAS for pain and the Japanese Knee Osteoarthritis Measure (JKOM) and WOMAC already a month after the injection, which remained at the same level at 6-months follow-up.

The platelet-rich plasma is proven to be a safe and effective method in managing knee OA. Marx suggested that the platelet density of PRP should be approximately four to five times that of whole blood to provide a sufficient platelet reserve for the release of various active biological factors [36]. The PRP used in this study complies with this standard. However, many authors emphasized the lack of standardization in terms of the PRP preparing procedure (speed and time of centrifugation), number of injections or intervals between injections. Belk et al. [11] reported that using a leukocyte-poor PRP should give better outcomes than a leukocyte-rich PRP; therefore, our intervention group received a leukocyte-poor PRP. We decided to perform three IA PRP injections at 7-day intervals according to Görmeli et al. [43], who reported that multiple PRP injections are significantly more effective than a single injection in patients with early OA. In our study, the patients suffered from mild to moderate OA and, therefore, were likely to benefit mostly from the applied IA PRP protocol.

To date, IA injections of AAT and PRP have not been compared in terms of the effectiveness of the treatment of knee OA. Only one study exists that compared these two methods, but in the treatment of the Achilles tendinopathy [44].

The performance of activities of daily living is crucial for patients who suffer from knee osteoarthritis. Therefore, it was a big surprise for us that the functional evaluation of knee OA patients after the biological treatment with AAT was included only in the Fodor and Paulseth [38] and Bansal et al. [45] studies, who tested TUG and 6 m Walk Distance (6 mWD), respectively. In both studies, a significant improvement in comparison to the baseline was found; however, Bansal et al. tested their subjects six times with the last follow up at 2 years after the AAT injection. There are no studies assessing the influence of IA injections of PRP on the functional capacity, but Altamura et al. [46] concentrated on the return to sport. They reported that, despite a significant improvement in the subjective scores (IKDC 2000, Tegner score, VAS Pain), only 76.6% of the participants returned to the sport activity and 48.9% returned at the same level of performance. In our protocol, we have assessed the quality of the performance of ADL of our participants, including standing up from a chair and walking on a flat surface.

There was no exercise program created for the patients from the intervention groups in our protocol. That is why we concluded that the reported improvement in MVIC came from the decrease in symptoms, which allowed the patients to generate a greater force during the tests. However, we observed that the strength did not change much at 1-month follow-up in the AAT group in comparison with the PRP group. This phenomenon might be caused by partial-weight bearing recommended after the AAT implantation procedure.

This study has some limitations. The first problem is a limited group of patients included into the statistical analysis. Moreover, the treated groups were not homogenous in gender. Second, there was no possibility for the participants to be blinded. The AAT procedure required hospitalization and general anesthesia; on the other hand, the IA injections of PRP were performed in an outpatient clinic. Third, there was no one, clear rehabilitation program for the participants. That is why physiotherapy intervention differed among the treated patients.

## 5. Conclusions

IA injection of AAT or PRP improves pain, symptoms, quality of life and functional capacity. Our results show comparable effectiveness in the treatment of patients with mild to moderate knee OA.

## Figures and Tables

**Figure 1 biomedicines-10-00684-f001:**
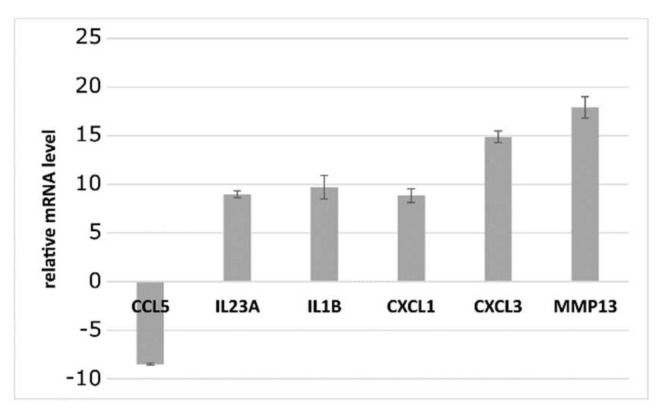
The relative expression levels of selected mRNAs in the platelet-rich plasma specimens. Data represent the expression levels of CCL5, IL23A, IL-1 β, CXCL1, CXCL3 and MMP13 in relation to GAPDH level, measured with the use of qRT-PCR. Data are represented as the mean ± standard deviation of twenty independent experiments.

**Figure 2 biomedicines-10-00684-f002:**
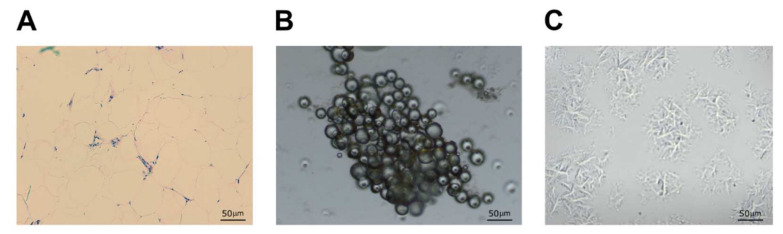
Autologous adipose tissue and adipocyte-derived mesenchymal cell morphology. Histological analysis of the adipose tissue (**A**) was observed under a light microscope. Nuclei are stained with hematoxylin (purple) and cytoplasmic components with eosin (pink). The morphology of the Lipogems product (day 1, (**B**)) and adipose-derived mesenchymal stem cells (day 28, (**C**)) was observed and photographed using a light microscope microscopy.

**Figure 3 biomedicines-10-00684-f003:**
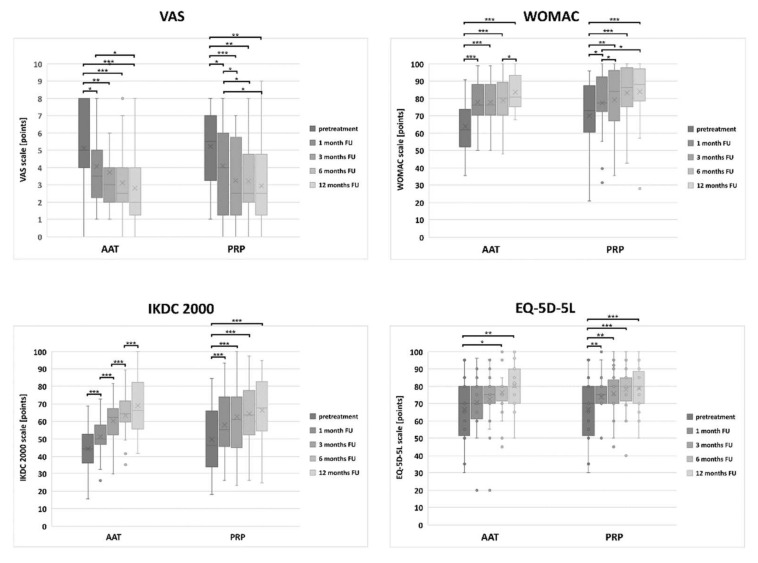
The distribution of the VAS, WOMAC, IKDC 2000 and EQ-5D-5L scores values in the PRP and the AAT treatment groups. Central lines represent the medians, boxes indicate the range from 25th to 75th percentile, whiskers extend 1.5 times above the interquartile range and outliers are represented as dots. The significance was designated as * *p* < 0.05, ** *p* < 0.01 and *** *p* < 0.001.

**Figure 4 biomedicines-10-00684-f004:**
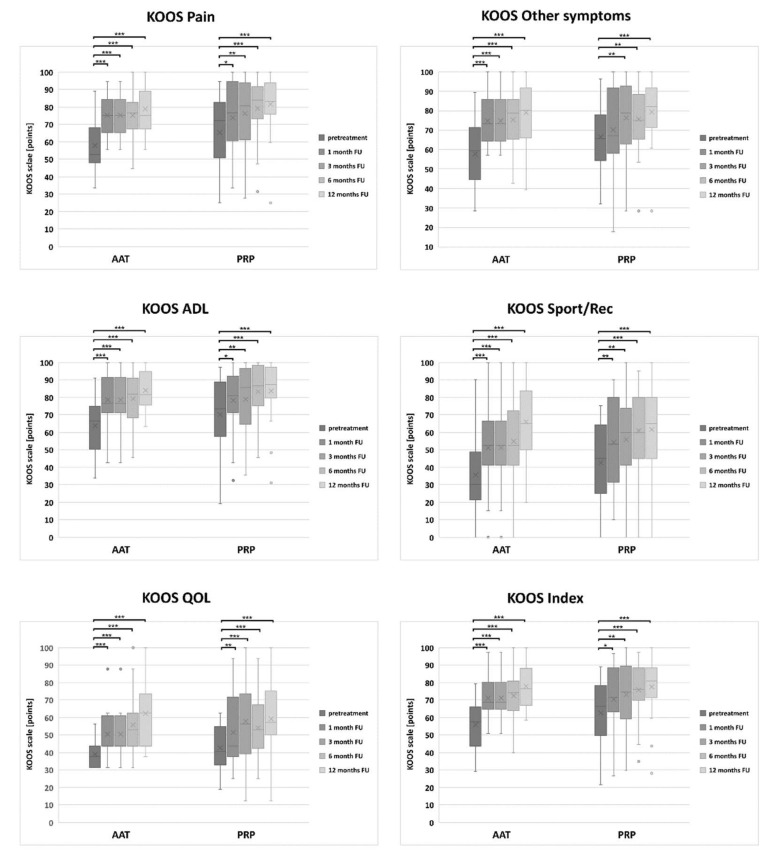
The distribution of the KOOS score values in the PRP and the AAT treatment groups. Central lines represent the medians, boxes indicate the range from 25th to 75th percentile, whiskers extend 1.5 times above the interquartile range and outliers are represented as dots. The significance was designated as * *p* < 0.05, ** *p* < 0.01 and *** *p* < 0.001.

**Figure 5 biomedicines-10-00684-f005:**
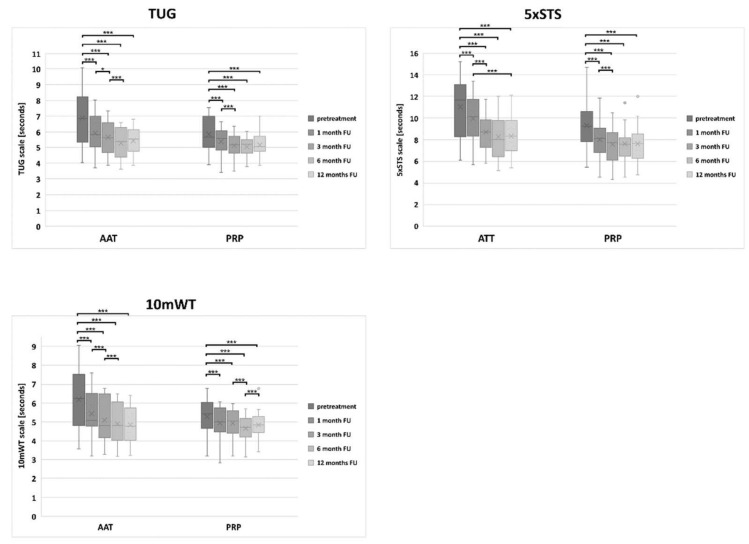
The distribution of the TUG, 5 × STS and 10 mWT score values in the PRP and the AAT treatment groups. Central lines represent the medians, boxes indicate the range from 25th to 75th percentile, whiskers extend 1.5 times above the interquartile range and outliers are represented as dots. The significance was designated as * *p* < 0.05 and *** *p* < 0.001.

**Figure 6 biomedicines-10-00684-f006:**
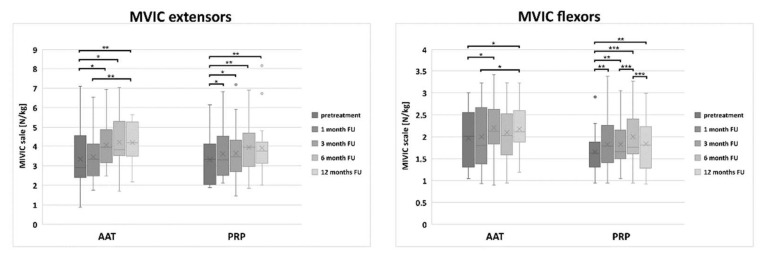
The distribution of the MVIC score values in the PRP and the AAT treatment groups. Central lines represent the medians, boxes indicate the range from 25th to 75th percentile, whiskers extend 1.5 times above the interquartile range and outliers are represented as dots. The significance was designated as * *p* < 0.05, ** *p* < 0.01 and *** *p* < 0.001.

## Data Availability

Data will be available from the corresponding author upon reasonable request.

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
