# Peer review of "Intra-Articular Injections of Autologous Adipose Tissue or Platelet-Rich Plasma Comparably Improve Clinical and Functional Outcomes in Patients with Knee Osteoarthritis"

_biomedicines, 2022, doi:10.3390/biomedicines10030684_

Round 1
Reviewer 1 Report
As author’s described, after AAT and PRP preparation, joint injection was processed. PRP injections were three times, and the interval was half a month. The ATT injection was processed just one time? If ATT injection also processed three times, authors need to describe more detail, including ATT processing, product storage, product safety test and injection volume.
Author Response
We appreciate the overall positive conclusion of the Reviewers and the possibility to improve our manuscript. Here we present a full list of changes.
Reviewer 1 - Comments and Suggestions for Authors
As author’s described, after AAT and PRP preparation, joint injection was processed. PRP injections were three times, and the interval was half a month. The ATT injection was processed just one time? If ATT injection also processed three times, authors need to describe more detail, including ATT processing, product storage, product safety test and injection volume.
Our response: The AAT injection was processed just one time, therefore there was no need for the storage of the product. We have clarified that in 2.5 section. Single injection of AAT for early knee OA has been studied in various clinical examinations, which encouraged us to follow the same protocol.
Reviewer 2 Report
Dear Authors: I want to congratulate with You for Your paper. Some informations are missing, and in my opinion these are important ones to be included to deserve publication in this journal. Also, some corrections are needed.
- randomization protocol /method should be described
- once an acronym is cited and use, always use it (PRP in line 391, for example)
- infos about Lipogem (house, place).
- why not to compare PRP and ATT to HA instead of healthy volunteers? this would be of great importance in my opinion: HA is cheaper and less invasive than PRP and even more than ATT.. You should stress some info about costs as well. also, why using healthy volunteers instead of osteoarthritic volunteer that were not undergoing any injection? these points could be discussed in discussion..
I recommend the paper to be accepted after minor revision.
Author Response
Reviewer 2 - Comments and Suggestions for Authors
Dear Authors: I want to congratulate with You for Your paper. Some informations are missing, and in my opinion these are important ones to be included to deserve publication in this journal. Also, some corrections are needed.
- randomization protocol /method should be described
Our response: We have added this missing information to the section 2.1, as suggested by the Reviewer.
- once an acronym is cited and use, always use it (PRP in line 391, for example)
Our response: We have changed that, as suggested.
- infos about Lipogem (house, place)
Our response: We have added this missing information.
- why not to compare PRP and ATT to HA instead of healthy volunteers? this would be of great importance in my opinion: HA is cheaper and less invasive than PRP and even more than ATT.. You should stress some info about costs as well. also, why using healthy volunteers instead of osteoarthritic volunteer that were not undergoing any injection? these points could be discussed in discussion.
Our response: The healthy volunteers group was an important part of the study, since it served for comparison of the input data, that is to make sure that the functional and clinical results of the OA patients are indeed deviate from the norms for healthy people. Though the HA treatment results would be scientifically interesting to compare with other biological treatments, we did not have the option to treat the patients with HA at the time of evaluations. Moreover, we strongly believe that creating a control group of osteoarthritic participants that are not undergoing any treatment for 12 months in unethical, therefore we did not include such group in the evaluations.
I recommend the paper to be accepted after minor revision.
Reviewer 3 Report
This is an interesting study evaluating the effect of AAT and PRP in patients affected by unilateral knee OA. The manuscript is well structured and the methodology supports the conclusion. However, the paper has some flaws that need to be addressed before considering submission:
- Abstract: According to Biomedicines' Instruction for Authors, the abstract should start with a brief introduction on the background of the study. In the section dedicated to methods, information about degree of knee OA, PRP preparation technique, AAT harvesting procedure as well as number and timing of injection should be provided. In addition, there is no explanation about in vitro assays mentioned here. Moreover, abbreviated forms (e.g., CCL5, TNF, KOOS etc.) should be preceded by their corresponding extended form when mentioned the first time. Same for other instances later in the text (e.g., line 54). Therefore, the abstract should be completely revised including these data while respecting the word limit.
- Extensive English language editing with the support of a native speaker is warmly advised. Improper use of articles, grammar, lexicon and syntax can be noted in several spots throughout the text.
- Please be consistent in the use of abbreviated forms throughout the text once introduced the first time.
- Lines 42-44: arthroscopic debridement for isolated knee OA is no longer supported by the latest guidelines. Ligament rupture is more often the the cause of knee OA rather than a consequence needing reconstruction surgery. Moreover, ligament reconstruction is not usually recommended in osteoarthritic knee joints. Therefore, the entire sentence should be reformulated. What do authors mean for cartilage reconstruction?
- Lines 71-78: Authors should revise their statements as several studies have recently evaluated the effectiveness of adipose-derived stem cell and/or PRP in the treatment of OA (10.3390/jcm11041056) and more specifically knee OA (10.1089/scd.2021.0053) focusing on pain and function outcomes.
- Detailed description of the randomization process as well as the organization of follow-up visits should be added to section 2.1. Why did healthy controls not undergo follow-up outcome evaluation like subjects in the experimental groups?
- I suggest to show Figure S1 directly in the manuscript rather than keeping it among the Supplementary Materials. Please renumber all other Figures accordingly.
- Lines 244-246: details about the action of cytokines and chemokines should be removed here and possibly elaborated in the Discussion section.
Author Response
Reviewer 3 - Comments and Suggestions for Authors
This is an interesting study evaluating the effect of AAT and PRP in patients affected by unilateral knee OA. The manuscript is well structured and the methodology supports the conclusion. However, the paper has some flaws that need to be addressed before considering submission:
Abstract: According to Biomedicines' Instruction for Authors, the abstract should start with a brief introduction on the background of the study. In the section dedicated to methods, information about degree of knee OA, PRP preparation technique, AAT harvesting procedure as well as number and timing of injection should be provided. In addition, there is no explanation about in vitro assays mentioned here. Moreover, abbreviated forms (e.g., CCL5, TNF, KOOS etc.) should be preceded by their corresponding extended form when mentioned the first time. Same for other instances later in the text (e.g., line 54). Therefore, the abstract should be completely revised including these data while respecting the word limit.
Our response: We have modified the abstract, as suggested by the Reviewer. We have introduced some missing information: the background, the information about the degree of knee OA, the AAT harvesting procedure. However, due to the word limit (max. 200 words) in the abstract section, we were not able to introduce all the changes suggested. We have therefore left the abbreviated forms of the genes and of the evaluation tests and explained them while first appearing in the main text.
Extensive English language editing with the support of a native speaker is warmly advised. Improper use of articles, grammar, lexicon and syntax can be noted in several spots throughout the text.
Our response: We have carefully went through the whole text and made some corrections. The manuscript has been edited by the native English speaker.
Please be consistent in the use of abbreviated forms throughout the text once introduced the first time.
Our response: We have corrected that, as suggested.
Lines 42-44: arthroscopic debridement for isolated knee OA is no longer supported by the latest guidelines. Ligament rupture is more often the the cause of knee OA rather than a consequence needing reconstruction surgery. Moreover, ligament reconstruction is not usually recommended in osteoarthritic knee joints. Therefore, the entire sentence should be reformulated. What do authors mean for cartilage reconstruction?
Our response: We are very thankful for this comment. We have changed this sentence accordingly.
Lines 71-78: Authors should revise their statements as several studies have recently evaluated the effectiveness of adipose-derived stem cell and/or PRP in the treatment of OA (10.3390/jcm11041056) and more specifically knee OA (10.1089/scd.2021.0053) focusing on pain and function outcomes.
Our response: We have revised our statement and changed the paragraph accordingly.
Detailed description of the randomization process as well as the organization of follow-up visits should be added to section 2.1. Why did healthy controls not undergo follow-up outcome evaluation like subjects in the experimental groups?
Our response: We have added the missing information concerning the randomization as well as the follow-ups to the section 2.1, as suggested by the Reviewer. The healthy controls did not undergo the follow-ups and were subjected to the evaluations just once, at the time “before the treatment”. The control group served in this study for comparison of the input data, that is to make sure that the functional and clinical results of the OA patients are indeed deviate from the norms for healthy people.
I suggest to show Figure S1 directly in the manuscript rather than keeping it among the Supplementary Materials. Please renumber all other Figures accordingly.
Our response: We have presented the figure in the main text as Figure 1, as suggested. The remaining figures were re-numbered accordingly.
Lines 244-246: details about the action of cytokines and chemokines should be removed here and possibly elaborated in the Discussion section.
Our response: We have changed that accordingly, as suggested.